# On-surface synthesis of enetriynes

Nan Cao [1,2,3], Biao Yang [1,3] ✉, Alexander Riss [1], Johanna Rosen[2], Jonas Björk [2] ✉ & Johannes V. Barth[1]

Belonging to the enyne family, enetriynes comprise a distinct electron-rich all-carbon bonding scheme. However, the lack of convenient synthesis protocols limits the associated application potential within, e.g., biochemistry and materials science. Herein we introduce a pathway for highly selective enetriyne formation via tetramerization of terminal alkynes on a Ag(100) surface. Taking advantage of a directing hydroxyl group, we steer molecular assembly and reaction processes on square lattices. Induced by $O_2$ exposure the terminal alkyne moieties deprotonate and organometallic *bis*-acetylide dimer arrays evolve. Upon subsequent thermal annealing tetrameric enetriyne-bridged compounds are generated in high yield, readily self-assembling into regular networks. We combine high-resolution scanning probe microscopy, X-ray photoelectron spectroscopy and density functional theory calculations to examine the structural features, bonding characteristics and the underlying reaction mechanism. Our study introduces an integrated strategy for the precise fabrication of functional enetriyne species, thus providing access to a distinct class of highly conjugated π-system compounds.

Enynes are conjugated π-bonded electron-rich compounds that are of broad interest due to their role in biochemistry and materials science[1,2]. The enyne family comprises four basic members with increasing number of substituted alkynyl groups, namely eneyne, enediyne, enetriyne, and enetetrayne (cf. Fig. 1a). The low conjugated eneyne and enediyne motifs have attracted much attention over the last decades since the recognition of their relevance for antitumor antibiotics[3,4]. Their biological activity stems from the generation of diradicals, which is responsible for DNA cleavage and cell destruction[5]. Beyond their anticancer prospects, enynes also represent elementary units in synthetic chemistry and materials science, which undergo transformations to build functional compounds and π-conjugated frameworks[6,7].

Motivated by the biological relevance and application potentials, many efficient strategies have been developed to afford enyne species. Common transition-metal-catalyzed cross-coupling reactions are generally based on complex materials such as vinyl halides[8], alkynyl halides[9] or organometallic alkenes[10], while oxidative cross/homo-coupling reactions apply to simple terminal alkynes[11], being more sustainable synthesis routes[1]. In practice, low-conjugated eneynes and enediynes are easily synthesized and successfully applied in the construction of numerous functional materials[2,12,13]. Moreover, there are many realizations of the seemingly complex enetetrayne[8,9,14,15]. By contrast, enetriynes have rarely been synthesized practically[16] except for an early exploration with low yield[17,18]. This drawback is likely due to the difficult oxidative addition of electron-rich enynes[19], limiting their application potentials as a highly conjugated π-system. Accordingly, the development of strategies to afford enetriyne motifs with high chemoselectivity represents a challenging endeavor.

On-surface synthesis[20,21] has introduced alternative routes towards the formation of conjugated nanostructures[22–24] and functional organic molecules[25–28]. The confinement and catalytic activity on surfaces promote the transformation of reactive groups, enabling the formation of distinct organic compounds or nanostructures[29,30]. The advent of bond-resolved non-contact atomic force microscopy (nc-AFM)[31,32] accelerated the precise fabrication of covalent organic nanoarchitectures[33–36]. Specifically, aryl-alkynes proved useful in a plethora of on-surface reactions[37,38], such as homo-coupling[39–44], cycloaddition[32,45–48], and cross-coupling with other functional groups[49–52]. The versatile chemical properties of terminal alkynes limit the control over the reaction pathways, inevitably entailing side products, incidentally mainly appearing in the form of enynes[53–57]. In

[1]Physics Department E20, Technical University of Munich, D-85748 Garching, Germany. [2]Department of Physics, Chemistry and Biology, IFM, Linköping University, 58183 Linköping, Sweden. [3]These authors contributed equally: Nan Cao, Biao Yang. ✉e-mail: biao.yang@tum.de; jonas.bjork@liu.se

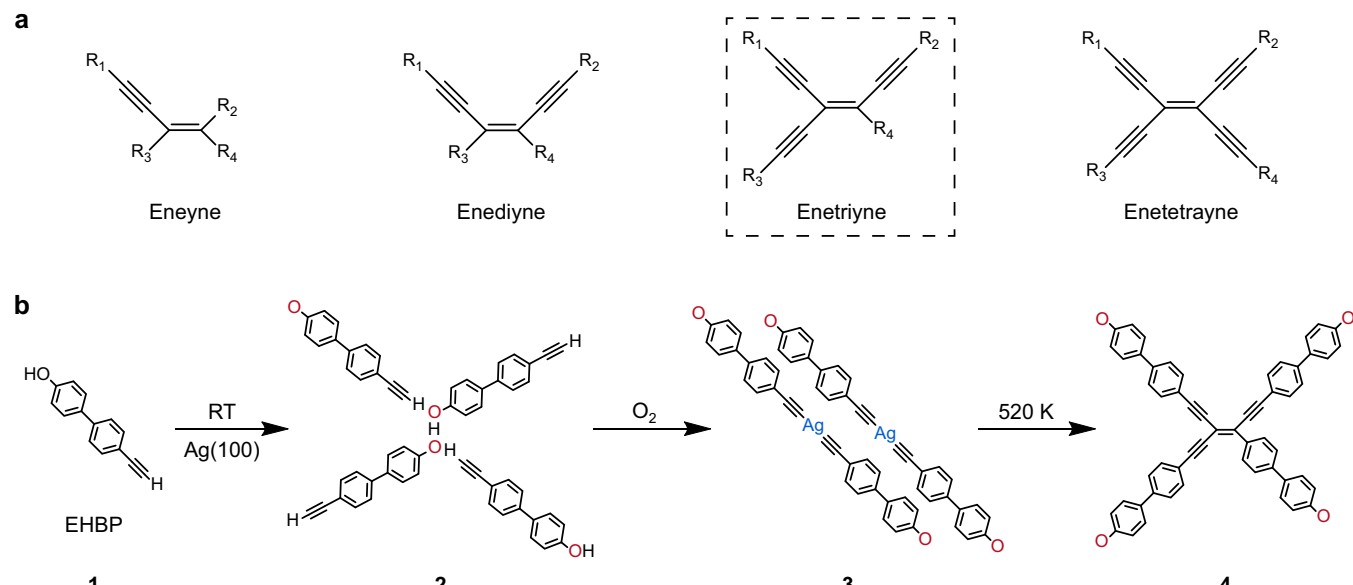

**Fig. 1 | Scheme of enynes and reaction pathway. a** The enyne family features elements with different electron-rich C-C bonding configurations. The marked enetriyne species is the main product obtained by the on-surface synthesis strategy developed in this work. **b** Schematic illustration showing the chemical transformations in this work. They undergo in stages pristine precursor 4′-ethynyl-[1,1′-biphenyl]−4-ol (EHBP) **1**, self-assembly structure **2**, organometallic dimer **3**, and covalent bonded enetriyne tetramer **4**, respectively. The devised multistep protocol involves a precursor with a directing group, gas-mediated pretreatment, and thermally activated reactions.

addition, the advantages of multicomponent precursors for the synthesis of complex surface structures have been recognized[58–61], though the full control of reaction pathway for each active group involved is notoriously difficult[41,49,61,62]. Steering reactions involving aryne modules has been improved via steric or template effects, which can inhibit undesired side reactions[40–43]. Furthermore, it is tempting to focus on enynes as targeted products by bestowing aryne precursors with a directing group, suppressing undesired pathways and creating spatial arrangements favoring the cross-addition reaction mechanism. Recently, Wang et al.[60] synthesized *cis*-enediynes on Ag(111) by introducing Br to a 4-ethynyl-1,1′-biphenyl precursor at either the phenyl side or the ethynyl end, whereby the release of Br adatoms upon adsorption drives *cis*-enediyne aggregation into close-packed islands and poses a high steric barrier to further reactions. Inspired by such advances, the question arises as to whether it is possible to synthesize higher conjugated enetriyne compounds from a simple *bis*-substituted precursor on the surface.

Here, we report an approach of enetriyne formation with high selectivity via tetramerization of terminal alkynes on Ag(100), studied by a combination of scanning tunneling microscopy (STM), nc-AFM and X-ray photoelectron spectroscopy (XPS) measurements, complemented by density functional theory (DFT) modeling. Moreover, the electronic properties of the enetriyne products were examined by scanning tunneling spectroscopy (STS) measurements and theoretical calculations. The use of *bis*-substituted precursor 4′-ethynyl-[1,1′-biphenyl]−4-ol, named EHBP (**1**, cf. Fig. 1b), containing an alkyne and a directing hydroxyl group allows to control the reaction process whereby a specific enetriyne species with high yield was obtained. This work demonstrates a multi-step synthesis strategy providing enetriyne molecules, whereby the molecular precursor **1** first self-assembles into regular networks stabilized by mixed hydrogen bonds (**2**, Fig. 1b). Subsequently, organometallic Ag-bis-acetylide dimers (**3**, Fig. 1b) are generated via exposure to $O_2$[63,64], that self-assemble into densely packed arrangements directed by the dehydrogenated hydroxyl endgroups[65], preventing from other reactions between the terminal alkynes. The embedded adjacent dimeric units are a favorable arrangement for the following tetramerization reaction during a further annealing step, affording predominantly conjugated enetriyne

cores (**4**, Fig. 1b). The dehydrogenated hydroxyl endgroup stabilizes the expression of tetrameric enetriyne regular assemblies, sterically hindering the surface mobility and further reaction of conjugated products. DFT calculations are performed to elucidate the observed arrangements and reaction pathways underlying the developed on-surface synthesis protocol.

## Results

### Supramolecular organic assemblies
We initially deposited EHBP molecules onto Ag(100) at low temperature (150 K), to prepare a sample with pristine precursors, since both terminal alkynyl[39,48,57] and hydroxyl[65–68] groups can be modified on silver surfaces at mild conditions. The STM data reproduced in Fig. 2a for a medium coverage (45% of a saturated monolayer) shows a local aggregation of molecules adsorbed on the flat terrace. They typically assemble into tetrameric cross-shaped supramolecular modules, as highlighted in the inset. A careful inspection reveals that the four-fold symmetric unit has two mirror planes perpendicular to the surface, in marked contrast to the chiral windmill-like structure usually formed by CH·π interactions between four ethynyl endgroups[47,57,60]. This appearance suggests that the tetramer is a result of intermolecular hydrogen bonding between the hydroxyl heads, as illustrated by the structural model in Fig. 2b. The chemical state of pristine monomers under the assembly conditions is supported by XPS (cf. Supplementary Fig. 1), whereby the O 1s spectrum shows a single peak centered at 533.5 eV, consistent with previously reported values for the adsorbed aromatic hydroxyl group[65–68].

Upon annealing the previously described sample to room temperature (RT), or by depositing EHBP on Ag(100) held at RT, extended and fully reticulated supramolecular networks evolve (cf. Supplementary Fig. 2a). When zooming into the assembly domain, STM observations (Fig. 2c) identify a rhombic unit cell. The high-resolution STM image (inset in Fig. 2c) clearly indicates a vertex with reduced symmetry as compared to the low-temperature modules. Each vertex connects four molecules, two of which approach each other in a head-to-head fashion, whereby the molecular axes are offset and the elongated tails of the other two precursors connect. Importantly, the XPS characterization of the RT deposition sample reveals a partial

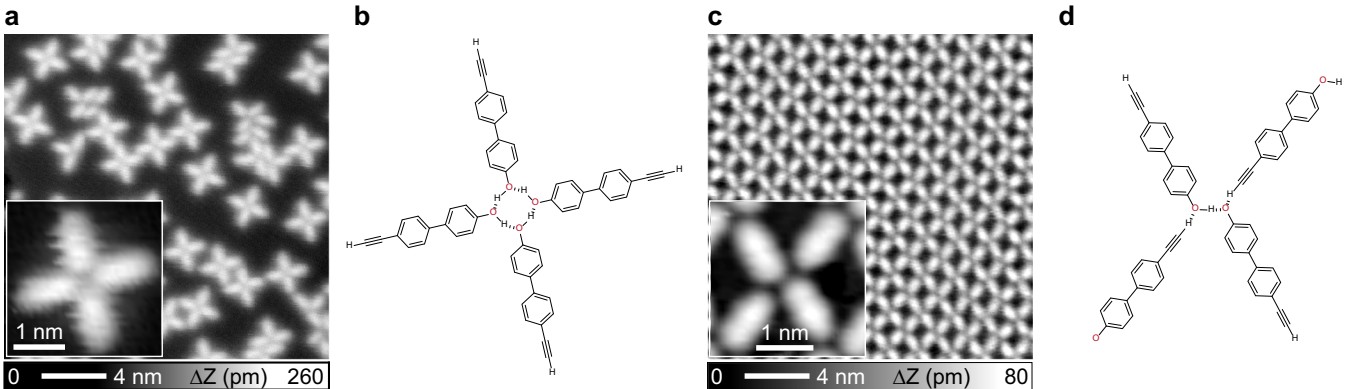

**Fig. 2 | Self-assembly structures of EHBP upon deposition on a clean Ag(100) substrate. a**, **c** With the substrate held at 150 K and room temperature, respectively, either tetrameric units (insets) or extended networks evolve. **b**, **d** The models for vertices occurring in (**a**) and (**c**) explain the respective structural arrangements stabilized by different hydrogen bonding schemes. Tunneling parameters: **a** and **b** $I_t = 1$ nA, $V_s = -100$ mV.

dehydrogenation of hydroxyl groups (see the "Discussion" below and also Supplementary Fig. 1), while the alkyne termini remain intact at RT in accordance to previous reports[57]. Consequently, a structural model for the supramolecular bonding motif is inferred, wherein C–H···O and O–H···O mixed hydrogen bonds form between alkyne and (dehydrogenated) hydroxyl moieties, as depicted in Fig. 1d. The marked change of molecular orientation (from Fig. 2b–d) is caused by the evolution of intermolecular interactions via chemical changes, since molecules exhibit sufficient mobility on the surface to aggregate into the respective preferred structures[60,65,68]. DFT modeling also supports the assignment of a structure with partially dehydrogenated constituents (see Supplementary Fig. 3).

### Organometallic dimers generated via O$_2$-exposure

Upon exposing the self-assembled networks to O$_2$ (~450 L) at RT, domains with regular stripe features in different orientations evolve (cf. Supplementary Fig. 2b). Figure 3a reveals two kinds of ribbons (16.8 Å *vs.* 21.4 Å wide) arranged alternately to form an extended domain. The repeated candy bar-like species are arranged parallel in both ribbons, the elements of which form an angle of 90°. The constituents are recognized as organometallic dimers composed of two EHBP monomers interconnected by an Ag adatom provided by the substrate. To better characterize this phase transition and the chemical nature of the linkages, bond-resolving nc-AFM data were obtained with CO-functionalized tips[69]. The AFM images in Figs. 3b and 3d reveal structural details of the organometallic dimer array regions marked as I and II in Fig. 3a, respectively. The biphenyl and alkynyl skeleton can be unambiguously distinguished as bright outlines. The central Ag adatoms are featureless (in contrast to their bright appearance in STM. These combined insights lead to the identification of alkynyl-Ag-alkynyl organometallic linked dimers, reminiscent of the alkynyl-silver bonding motifs on Ag(111) obtained in a similar manner by an O$_2$-mediated synthesis protocol[63,64]. Comparatively, both end groups of the dimers exhibit a dark appearance, similar to the nc-AFM resolution of ketone groups[70,71]. In addition, previous studies have reported that O$_2$ exposure may induce dehydrogenation of hydroxyl groups on silver[72,73]. We thus hypothesize that the organometallic dimers have both hydroxyl groups dehydrogenated, which assignment is corroborated by XPS measurements (*vide infra*), thus driving dimers arrangement in extended array domains through the hydrogen bonds between oxygen termini and biphenyl backbones.

It is interesting to note that dimers in array II (Fig. 3d) typically show two-fold symmetry, while those in array I (Fig. 3b) have reduced symmetry, which is mainly reflected in the bending of the molecular linkers. Another distinguishing feature is the different interplay in the vicinity of hydroxyl endgroups. The former is facing the entral part of

the diphenyl backbone of the dimer in array I (indicated by the dashed circle in Fig. 3d), while the latter points to a phenyl moiety of the dimer in array II (indicated by the dashed circle in Fig. 3b). To rationalize this stripe configuration, DFT calculations were performed for the organometallic ribbon phase. The optimized adsorption model of an organometallic dimer indicates that Ag adatoms reside on the four-fold Ag(100) hollow sites (Supplementary Fig. 4), for which the simulated STM images mimic the experimental appearance. Furthermore, the model in Fig. 3c shows the energetically most favorable configuration of the array structures. All Ag adatoms adsorb in hollow sites, and both terminal oxygens as well as alkynyl-Ag-alkynyl organometallic species match well with the two adjacent arrays in the AFM image (Fig. 3e). Also additional comparisons of simulated and experimental bias-dependent STM images agree nicely (Supplementary Fig. 5). Besides these regular arrangements of individual organometallic dimers, some interesting pairs of dimers emerge in array II, as shown in the STM and AFM images in Supplementary Fig. 6. Note that there is an obvious increased proportion of pair-dimers with less O$_2$ dosage. This is ascribed mainly to a different interplay in the vicinity of terminal oxygen, presumably due to incomplete dehydrogenation of hydroxyl groups (see the detailed XPS analysis below).

### Enetriyne formation via thermally activated tetramerization reaction

Upon further annealing the organometallic ribbon structure to 520 K, a phase evolution was observed. Overview STM (Supplementary Fig. 2c) indicate a regular superlattice topography, i.e., the anisotropic organometallic dimer-array patches transformed into a large-scale isotropic pattern. Figure 4a displays a STM image of the obtained ordered islands, revealing a self-assembled arrangement of newly formed cross-shape tetrameric species. For a better inspection of these products, a high-resolution STM image of a tetramer unit is depicted in Fig. 4b. The latter consists of four cross-connected rod-like parts, which are readily assigned to four EHBP derivatives. Note that in all STM images there are no indications of embedded Ag atoms from the seamless contours, which strongly suggests covalent carbon–carbon linkage at the crosses' centers. This also clearly differs from the supramolecular tetramers described above with different H-bonding nodes (Fig. 2a, b).

nc-AFM measurements provide further structural details and the carbon scaffold of the products. Figure 4c shows AFM data acquired at the same area of Fig. 4b imaged by STM. The bond-resolved inspection clearly reveals that four biphenyl backbones are covalently connected at the center, i.e., there is a tetra-substituted skeleton. When focusing on the central moiety, three of the linkages (between phenyls and the central connection) display bright line-shaped features, unambiguously recognized as ethynyl groups based on comparisons with earlier

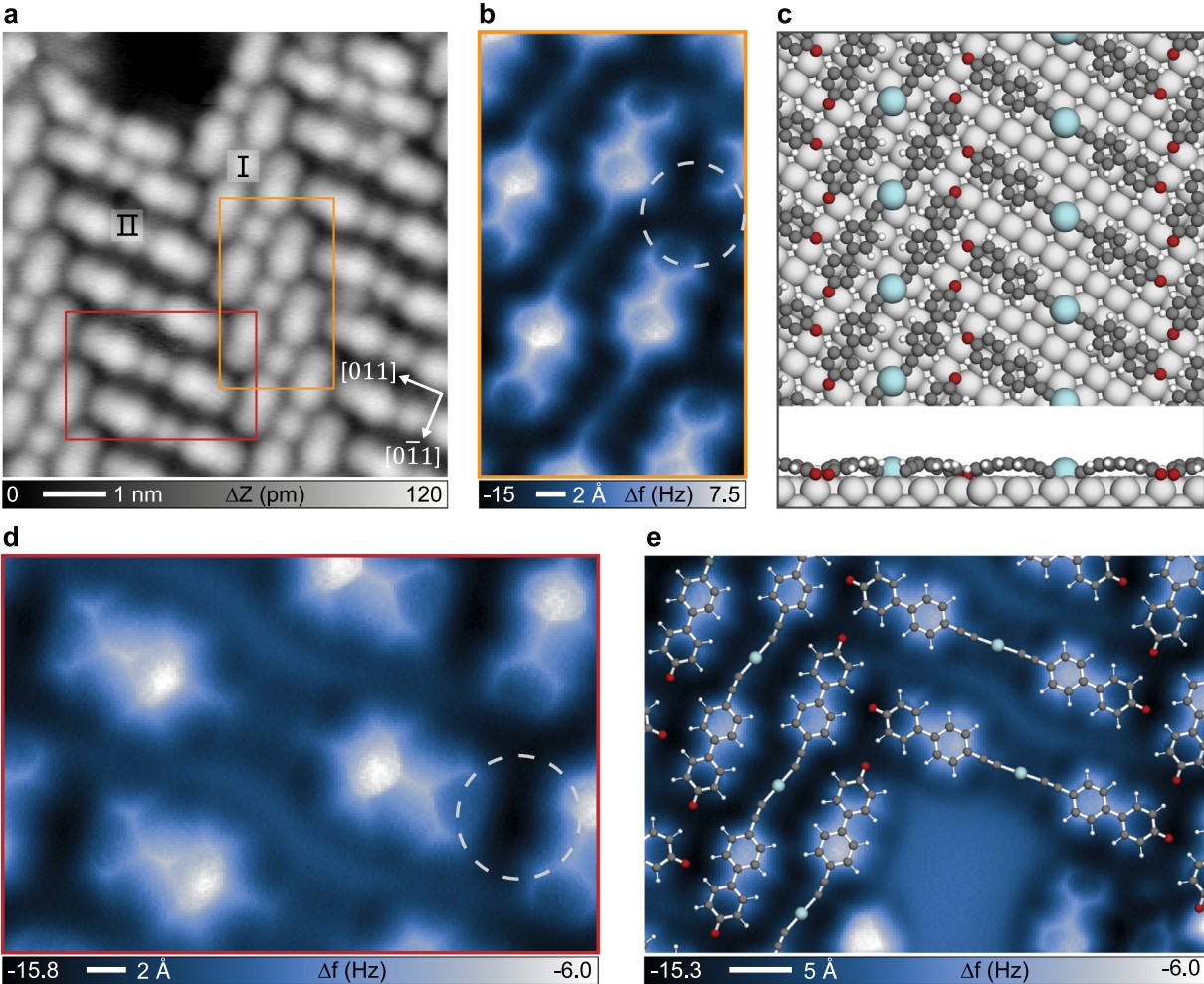

**Fig. 3 | Formation of organometallic ribbon phase via $O_2$ mediated on-surface reactions. a** Representative STM image revealing dimeric organometallic constituents and their mutual arrangement. **b**, **d** Bond-resolved nc-AFM images of the dimers corresponding to the area marked with orange and red rectangles in (**a**), respectively. The dashed circles indicate the intermolecular interaction modes between adjacent arrays. (**c**) DFT optimized organometallic structures on Ag(100). (**e**) AFM image of the organometallic structure partially superimposed with the optimized structural model in (**c**). Measurement parameters: **a** $I_t = 10$ pA, $V_s = 100$ mV; **b**, **d**, and **e** $V_s = 0$ V, constant height. Source data are provided as a Source data file.

AFM observations using analogously a CO tip[32,48]. Furthermore, a relatively short bond connects the last substituent and the central moiety, implying that the central bond derives from the fourth constituent. Considering its appearance as phenyl ring and the planar property of the entire tetramer, we confidently identify the tetramer as an enetriyne derivative (**4**, Fig. 2b).

Note that the phenyl rings in the periphery of the enetriyne tetrameric species appear darker and less sharp, due to a reduced adsorption height, which results from the strong interaction between terminal oxygen atoms and the substrate[70,71]. This is also supported by DFT calculations: Fig. 4d shows the most stable adsorption configuration of an enetriyne tetramer on Ag(100), whereby the molecular structural model resembles the shape of the observed tetramer products. The simulated AFM image (Fig. 4e) acquired with the DFT-optimized structure compares well with its experimental counterpart (Fig. 4c). Note that all oxygen atoms adsorb on bridge sites and bend the connected phenyl rings downward, in agreement with the AFM appearance. Hereby the oxygen atoms exist in the form of ketones engaging in intermolecular interaction with surrounding tetramers, which are observed in XPS as discussed below. A closer inspection of the adjacent arrangement of two tetramer products clarifies the importance of the supramolecular interactions for the formation of the well-ordered array (Supplementary Fig. 7).

Moreover, the non-four-fold symmetric nature of enetriynes inevitably entails variable adsorption orientations. Figure 4f, g shows STM and AFM images of a small area including four tetramer products. While all oxygen atoms face biphenyl center of the adjacent tetramer in the same way, the orientation of the enetriyne core varies between neighboring tetramers as the overlaid models illustrate. However, except for the slight difference in the length of the biphenyl skeleton there are no morphological differences neither from experimental nor simulated STM images (Supplementary Fig. 8), in agreement with its highly conjugated electronic structure. Importantly, although there are some molecular side products or spurious elements, such as trimers or deformed tetramers (Supplementary Fig. 9), the chemoselectivity towards the enetriyne is high (~85%), as deduced from a statistical analysis of extended STM data sets (Supplementary Fig. 10). Note that side products are mostly found at the edges of the domains. Hence we may obtain even higher yields by increasing the molecular coverage and $O_2$ dosage.

### XPS characterization

The varying chemical state of oxygen at different reaction stages is monitored with XPS (Fig. 5). Due to the partial dehydrogenation of hydroxyl groups at RT (Supplementary Fig. 1), the as-deposited sample at RT already comprises coexisting oxygen species. The O 1s

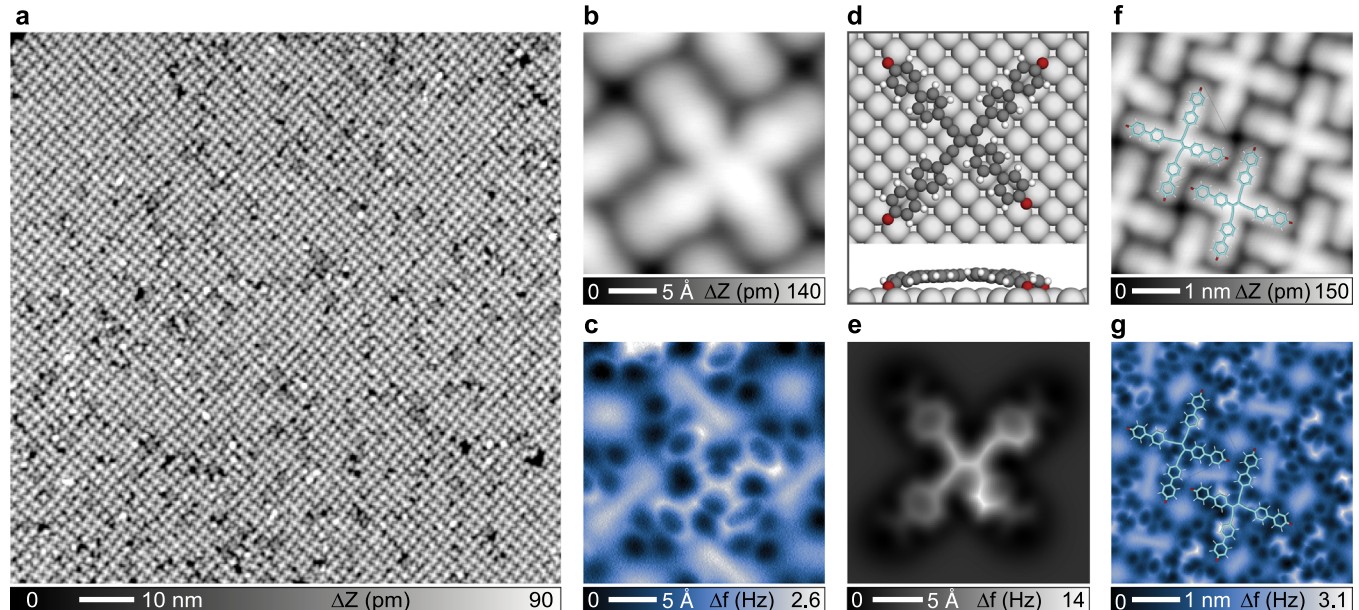

**Fig. 4 | Selective on-surface synthesis of enetriynes upon annealing the organometallic arrays at 520 K. a** Representative large-scale STM image of tetramer superlattice. **b** High-resolution STM image of a tetramer unit. **c** The corresponding bond-resolved AFM characterization reveals the carbon backbone. **d** DFT calculated structural model of an enetriyne species adsorbed on Ag(100). **e** Simulated AFM image of an enetriyne species. **f, g** STM and AFM images of an area of regular enetriyne arrays with partially overlaid structural models. Measurement parameters: **a** $I_t = 10$ pA, $V_s = 200$ mV; **b** and **f** $I_t = 10$ pA, $V_s = 100$ mV; **c** and **g** $V_s = 0$ V, constant height. Source data are provided as a Source data file.

spectrum in the upper panel of Fig. 5 displays two comparable peaks at binding energy (BE) of 532.2 eV and 530.7 eV, which are attributed to -C–OH and -C–O species, respectively[65–68]. This chemical evidence confirms the proposed model of the RT assembly structure shown in Fig. 2c, d. After O₂ exposure treatment, there is no evidence of further oxygen uptake at the surface, and the EHBP O 1s signature shown in the central panel of Fig. 5 reveals a pronounced reduction of the higher BE contribution (532.2 eV) along with an increase of the lower BE peak (530.4 eV). This change is in agreement with a substantial proportion of dehydrogenated O endgroups in the organometallic dimer arrays. Hereby the small contribution of –C–OH units (532.2 eV) is associated with the pairwise dimers in array II (cf. Fig. 3a and Supplementary Fig. 6), explaining the different interplay in the vicinity of terminal oxygen. Post-annealing the O₂-treated sample at 520 K leads to a single O 1s peak at 530.2 eV, indicating prevalence of enetriyne tetramers with exclusively -C–O endgroups. Note that the BE of the oxygen shifts slightly during the two-step treatment, which is presumably related to the modified intermolecular interactions in the different assemblies. In addition, C 1s spectra obtained at each stage show a downward shift, indicating extended conjugation properties (Supplementary Fig. 11). Therefore, the combined insights from the STM analysis, DFT modeling, and XPS characterizations, as well as bond-resolved AFM data, provides both convincing chemical and structural evidence for the on-surface synthesis of enetriyne tetramers.

## Reaction mechanism

In addition to the validation of synthesized products, it is also important to disentangle the pathway of the chemical reaction. The identification of intermediate states and incomplete products is an effective strategy to deduce the reaction process. The real-space structure analysis of STM is advantageous to study on-surface reactions and can capture trace intermediate state or byproducts for further analysis. With the help of STM and AFM observations, we found intermediate states in both steps of the synthetic process. Whereas the first stage of O₂-mediated organometallic coupling is well understood

from previous studies[63,64,72,73], the presently encountered second stage of thermally triggered tetramerization requires a detailed scrutiny. After annealing the organometallic ribbon structure to 450 K, we found initial signs of the reaction at the edge of the domains, and in the interior (typically of region II) broken dimers, irregular trimeric and tetrameric species (cf. Supplementary Fig. 12). Some of these entities can be assigned to intermediate states of the tetramerization reaction, which might involve stepwise organometallic bond scission and associated addition reactions.

Thus two possible addition pathways (Supplementary Fig. 13) are envisioned underlying the enetriyne formation, depending on the chemical structure of the monomers after deconstruction of organometallic dimers. To rationalize the chemical conversions, we performed a DFT-based transition state theory analysis. Firstly, the calculations consider the extraction of the Ag adatom from an organometallic dimer. The modeling indicates that the kinetically most accessible pathway proceeds with the Ag adatom fully released from the dimer, with an energy barrier of 1.03 eV (Supplementary Fig. 14). The resulting monomers have active -C≡C heads for further addition to the vicinal triple bond of organometallic dimers, indicating that reaction pathway 1 (Supplementary Fig. 13) is preferred. Based on these results, and in order to approach the enetriyne formation in a comprehensible manner, we assumed an initial state for the tetramerization with one organometallic dimer and two adjacent free monomers connected to the surface. The most favorable energy pathway posing such an initial state was identified by systematic computational modeling and is depicted in Fig. 6. The addition reaction is triggered by the Ag adatom abstraction from organometallic dimers. Subsequently, the enetriyne species are formed through a three-step addition reaction. The first addition reaction proceeds through an EHBP entity added into an organometallic dimer (IS to InS2) with an energy barrier of 1.07 eV (an almost equal energy barrier of 1.08 eV was obtained for the alternative pathway, shown in Supplementary Fig. 15). This process is followed by the addition of the second EHBP entity to the activated carbon of the trimeric species in InS2 (InS2 to InS3), with an energy barrier of 0.90 eV. The last step is the dissociation of the second Ag

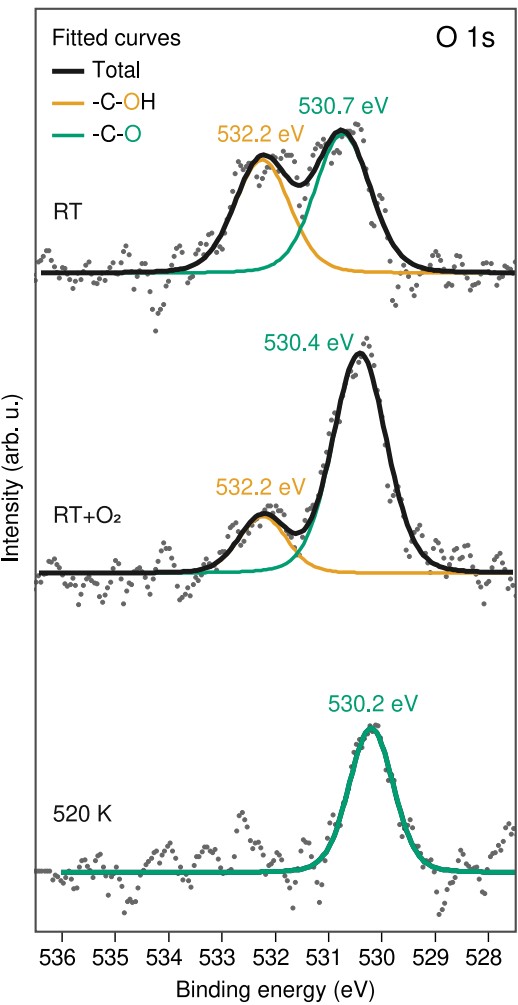

**Fig. 5 | XPS of the EHBP O 1s core level at different reaction stages after deposition on clean Ag(100).** The O 1s signature for the arrays formed at RT display two peaks at 532.2 and 530.7 eV with almost equal intensities, indicating the partial dehydrogenation of hydroxyl entities. The peak at 532.2 eV vanishes after annealing the substrate at 520 K, revealing fully dehydrogenated hydroxyl end-groups. Gray dots in the plots denote the linear-background-subtracted XPS data, and all peaks were fitted using a Lorentzian line shape, LA(1.53,243). Source data are provided as a Source data file.

adatom and the addition of the third EHBP entity, which requires a large energy barrier of 1.41 eV (InS3 to FS). Essentially, the barrier becomes slightly larger (0.11 eV) when including vibrational entropy and enthalpy (520 K) (Supplementary Fig. 16), while remaining small enough to allow for the identified coupling reaction at the experimental conditions[74]. The last step of the reaction is the rate-limiting one, but the intermediate state IntS3 has been rarely observed experimentally (Supplementary Fig. 12). However, the last step of the reaction is highly exothermic (IntS2 to IntS3). Thus, there will be excess energy in the system at IntS3 which could facilitate an efficient crossing of the final barrier. A similar phenomenon was shown for the homo-coupling between terminal alkynes on Ag(111)[75].

## Discussion

As suggested by the DFT calculations and multitechnique experimental analysis described above, the identified reaction mechanism is based on the alternative dimer arrays, which are derived not only from the O₂ treatment but also from the directing effect of both the terminal oxygen and the Ag(100) surface atomic lattice. Particularly, the hydroxyl groups play a crucial role in tuning interactions with adjacent

molecules and the substrates and thus substantially increase the high selectivity of the reaction. This is firmly demonstrated by the diversity of products when applying the same reaction trigger strategy on aryl-alkyne precursors (4-ethynylbiphenyl (EBP) and 4,4′-diethynylbiphenyl (DEBP)) without the directing hydroxyl groups (Supplementary Fig. 17 and Supplementary Fig. 18). To gain insights into the control parameters of the high selectivity of the formation of enetriynes with EHBP, we performed comparative experiments without O₂ dosage step and/or on Ag(111) surface. As expected, direct annealing of the RT deposited EHBP/Ag(100) sample (~45% of a full monolayer) up to 420 K results in disordered products (Supplementary Fig. 19). More interestingly, while the O₂ dosage on EHBP/Ag(111) sample induces the deprotonation of alkynyl groups and the formation of organometallic dimers, the condensed aggregation structures cannot afford effective control for the thermally triggered addition reaction upon further annealing (Supplementary Fig. 20). Therefore, the square symmetry of Ag(100) surface lattice plays an important role in the aggregation of organometallic dimers and subsequently evolving enetriyne species. These results again evidence our effective control of the enetriyne on-surface synthesis.

As highly conjugated electron-rich compounds, enetriynes are associated with promising electronic properties. Theoretical calculation for the density of states (DOS) were performed based on the optimized structural model of an enetriyne species adsorbed on Ag(100) (cf. Fig. 4d). We plotted the projected DOS (PDOS) and distinguished multiple pronounced orbitals which are attributed to the separate moieties of the molecular structure. As shown in Fig. 7a, the most noticeable peak centered at 0.5 V represents the contribution from the enetriyne junction. With the conjugation of enetriyne in between, the HOMO-LUMO gap of the aromatic phenyl backbones is close to ~2.0 eV. To get insight into the local electronic properties of the enetriyne tetrameric units, we performed STS differential conductance measurements (dI/dV) on the center of a tetramer (Fig. 7b), and identified one predominant peak (−2.1 V) within a large energy window. With a closer inspection of the [−1.8 V, 1.2 V] bias range, several peaks referring to molecular orbitals are disentangled as compared to the reference spectrum of clean Ag(100) (Fig. 7c). Without observing the state near 0 V, a derived energy gap of 1.8 eV is close to the calculated gap of aromatic backbones. The reasons for this might be of the large energy gap of the biphenyl backbones through connection to the conjugated enetriyne cores, as well as electronic hybridization with the substrate.

In conclusion, we present an approach to synthesize well-defined enetriyne species with a high yield (~85%) on the Ag(100) surface. The selectivity of the surface-confined reaction was achieved by introducing (i) a hydroxyl group to direct the arrangement in the supramolecular domains, (ii) O₂-mediated conversion of ethynyl groups to afford organometallic species while avoiding the homo-coupling, and (iii) the square-lattice Ag(100) substrate directing the addition reaction to form tetrameric enetriynes. This result is confirmed by a multi-technique comprehensive experimental examination. With these systematic observations molecular arrangements and chemical conversions were identified, suggesting a reaction pathway that was systematically investigated by DFT calculations. The electronic properties of the enetriyne tetramers were studied by STS measurements and PDOS calculations, signaling a high conjugation of the enetriyne junction and connected aromatic ring systems. The present integrated synthetic approach notably combines a general precursor design and customized experimental preparation protocols based on the recognized properties of molecular substituents as well as the substrate symmetry and catalytic activity. Our study introduces a high-yield procedure for the in vacuo preparation of enetriyne derivatives on surfaces. We envision that the gained knowledge and methodological insights may be further developed towards the precise synthesis of molecular wires or reticulated two-dimensional networks featuring

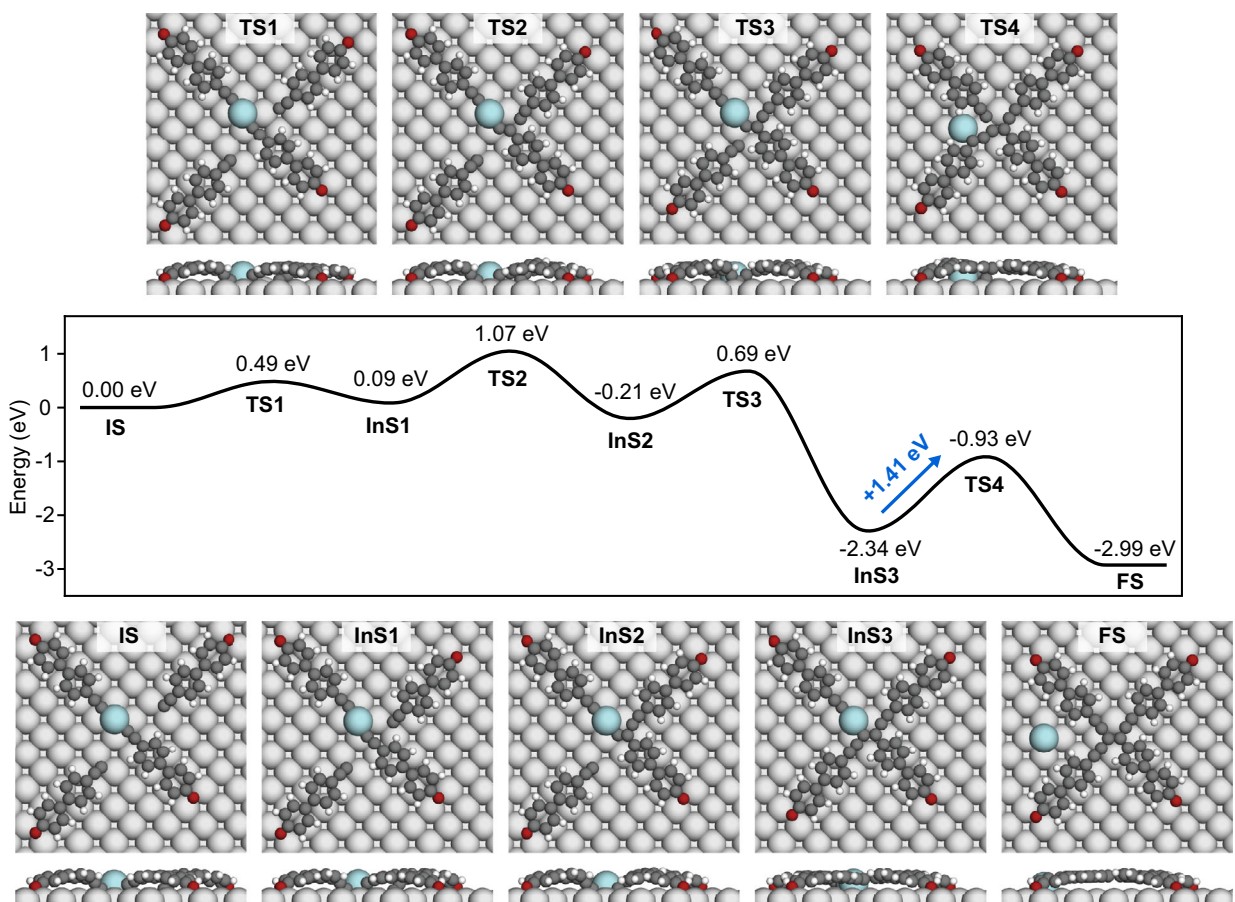

**Fig. 6 | Reaction pathway and energy profile from organometallic dimers toward the formation of enetriyne derivatives on Ag(100).** C, H, O, surface Ag atom, and Ag adatom are represented by the dark gray, white, red, light gray, and cyan balls, respectively. Source data are provided as a Source data file.

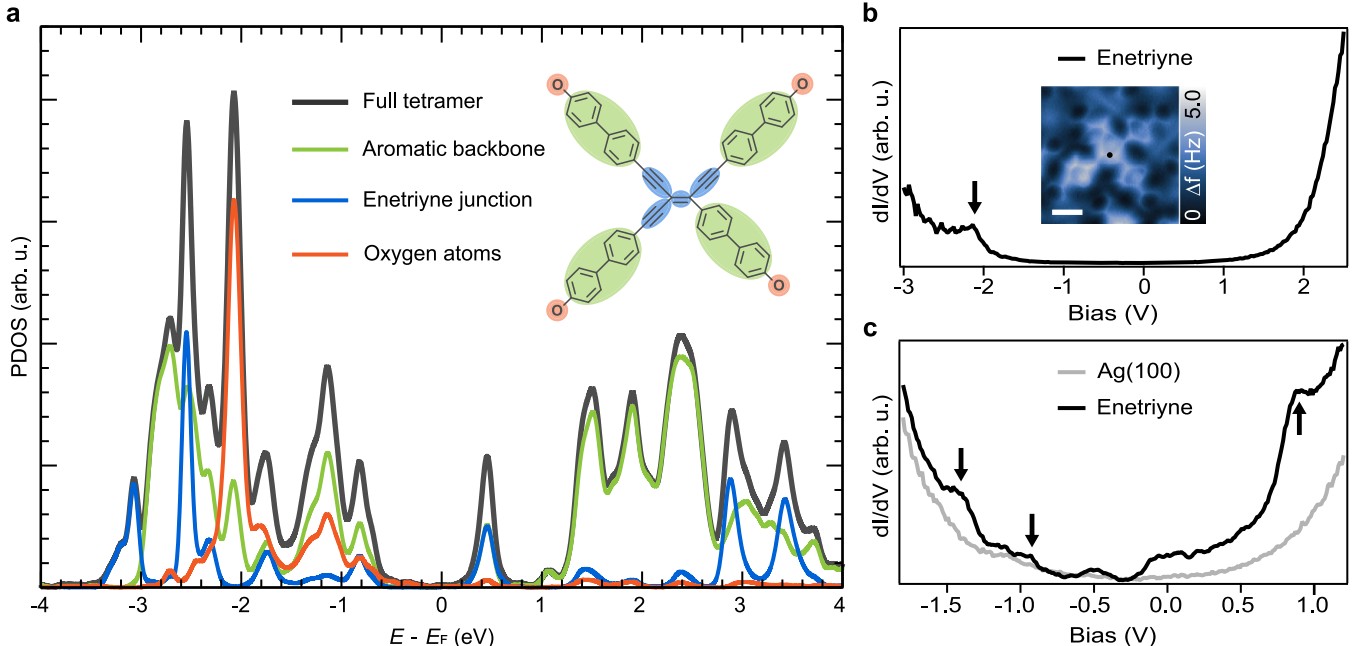

**Fig. 7 | Electronic structure of the enetriyne tetramer products. a** Theoretical PDOS calculated for an enetriyne tetramer adsorbed on a Ag(100) surface. The black, green, blue, and orange solid lines represent the DOS projected on the full tetramer, aromatic backbone, enetriyne junction, and oxygen atoms, respectively. **b** Differential conductance dI/dV spectrum (−3 V - +2.5 V) taken at the center of a

tetramer as indicated by the black point on the inset AFM image. Scale bar is 5 Å. **c** dI/dV spectrum (−1.8 V - +1.2 V) acquired on the enetriyne motif and the reference spectrum acquired on the atomic clean Ag(100) surface (gray solid line). Source data are provided as a Source data file.

embedded enetriyne units, affording additional functionalities and bearing application potential.

## Methods

### Sample preparation
The Ag(100) and Ag(111) single crystal substrates were cleaned via cycles of $Ar^+$ sputtering (1.0 kV) and annealing at 720 K. The precursor EHBP was purchased from AmBeed Company with a normal purity of >95%, EBP and DEBP were purchased from TCI company. Molecules were deposited from quartz crucibles onto the substrates held at room temperature or 150 K. The deposition time was appropriately controlled to obtain a molecular coverage of ~45% of a saturated monolayer. The annealing time is 10 minutes for each reaction step.

### STM/STS and AFM measurements
Experiments have been carried out in two separate ultrahigh vacuum systems. Initial STM measurements were performed with a commercial Joule-Thomson-STM (SPECS) at 4.5 K (base pressure below $2.0 \times 10^{-11}$ mbar), the corresponding STM images were taken in constant current mode using a tungsten tip, and the bias voltage was applied to the sample. The scanning parameters (tunneling current $I_t$ and sample bias $V_s$) are given in the respective figure captions. The STS dI/dV spectra were acquired with a lock-in amplifier (frequency $f = 954$ Hz, and modulation $V_{rms} = 20$ mV). Subsequent STM/AFM measurements were performed with a commercial instrument (CreaTec) at 5 K (base pressure below $4 \times 10^{-10}$ mbar). These STM images were also recorded in constant current mode. Bond-resolved AFM measurements were acquired at constant heights and $V_s = 0$ V using a qPlus tuning fork sensor[76] (resonance frequency ≈31 KHz, oscillation amplitude 60 pm, Q value > 100,000, stiffness $k ≈ 1800$ N m$^{-1}$) operated in frequency modulation mode. The measurements were performed with CO-terminated tips[31,77], obtained by vertical manipulation of adsorbed CO molecules that were dosed onto the substrate at T < 10 K. The data were analyzed using WSXM[78] and SpmImage Tycoon[79].

### XPS measurements
XPS measurements were performed in a SPECS GmbH UHV system (base pressure of $3 \times 10^{-10}$ mbar). A XR50 X-ray source with ellipsoidal crystal FOCUS 500 monochromator provided monochromatic Al K$\alpha$ radiation ($h\nu = 1486.71$ eV). Spectra were recorded with a PHOIBOS 150 hemispherical analyzer in normal emission geometry with the samples held at 300 K. The binding energy of all spectra was calibrated against the Ag $3d_{5/2}$ core level of the silver at 368.3 eV. For XPS spectra analysis, a Shirley (C 1s) or linear (O 1s) baseline were employed for background subtraction from the raw data, and all spectra were fitted using a Lorentzian line shape, LA(1.53,243) in CasaXPS software.

### DFT calculations
The calculations were performed in the DFT framework using the Vienna ab-initio simulation package (VASP)[80]. The projector-augmented wave (PAW) method was used to describe the interactions between ions and electrons[81]. The exchange-correlation interactions were treated by van der Waals density functional (vdWDF)[82] in the version of rev-vdWDF2 proposed by Hamada[83], which has shown to accurately describe adsorption heights for molecules adsorbed on Ag surfaces[84,85]. The transition states were searched by a combination of Climbing Image Nudge Elastic Band method and Dimer method[86–88]. Firstly, 10 images were inserted in between the initial and final states. The central images were further used as the input of the Dimer calculations in order to obtain precise transition states. Plane waves were used as a basis set with an energy cut-off of 400 eV. In all calculations, the Ag(100) substrate was modeled by four-layered slabs where the bottom two layers were fixed, and the periodic image interactions were avoided by implying a 15 Å vacuum region. The atomic structures were relaxed until the energy was less than $10^{-6}$ eV and the residual forces on all unconstrained atoms were less than 0.01 eV/Å. Surface unit cells of $p(13 \times 13)$ for the tetramer model, $p\left(\begin{smallmatrix} 9 & 3 \\ 3 & -8 \end{smallmatrix}\right)$ for self-assembled enetriynes, and $p\left(\begin{smallmatrix} 9 & 10 \\ -3 & 2 \end{smallmatrix}\right)$ for self-assembled organometallic dimers were used, and the 1st Brillouin zone was sampled by the gamma point only. In addition, the surface unit cell of $p(14 \times 6)$ is used for the organometallic dimer model, together with a $1 \times 2$ k-point sampling. The density of states of the surface-supported enetriyne tetramer was calculated using a k-point mesh $(3 \times 3 \times 1)$ in combination with the tetrahedrom method with Blöchl corrections[81]. The STM simulation images were calculated based on the Tersoff-Hamann approximation[89], AFM simulation images based on the geometries optimized by DFT were obtained on website: http://ppr.fyu.cz/. The Probe Particle Model is used in the simulation using classical force-fields[90,91].

### Reporting summary
Further information on research design is available in the Nature Portfolio Reporting Summary linked to this article.

## Data availability
All data needed to evaluate the conclusions in the paper are present in the paper and/or the Supplementary Information. Source data are provided as a Source data file. Source data are provided with this paper.

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

## Acknowledgements

This work was supported by funding from the Deutsche Forschungsgemeinschaft (German Research Foundation, DFG) under Germany's Excellence Strategy—EXC 2089/1 – 390776260 (e-conversion). N.C. was a holder of China Scholarship Council (CSC) PhD stipend. B.Y. acknowledges the Alexander von Humboldt-Foundation for a Research Fellowship for Postdoctoral Researchers. J.B. held a TUM Visiting Professorship (supported by the Bavarian Ministry of Science, Research and Arts). J.B. and J.R. acknowledge funding from the Swedish Research Council. Computational resources were allocated by the Swedish National Infrastructure for Computing and carried out at the National Supercomputer Centre, Sweden. A.R. acknowledges funding by the DFG project 453903355.

## Author contributions

B.Y. and J.V.B. conceived and supervised the experimental study. N.C. and B.Y. performed the STM, AFM, and XPS measurements. J.R. and J.B. supervised the theory study and N.C. performed the DFT calculations. N.C., B.Y., A.R., and J.V.B. analyzed the data and co-wrote the paper. All authors discussed the results and commented on the manuscript.

## Funding

## Competing interests

The authors declare no competing interests.
