## [Peer Review File · Nature Communications]

On-surface synthesis of enetriynesREVIEWER COMMENTS

Reviewer #1 (Remarks to the Author):

Cao and coauthors reported the selective synthesis of an enetriyne on Ag(100) by combined STM, AFM, XPS and DFT studies. The authors claim that the high selectivity stems from the introduction of a directing hydroxyl group, the formation of the alkynyl-silver-alkynyl organometallic intermediate species facilitated by the dosage of O₂, and the directing effect of the Ag(100) substrate lattice as well. The identification of the molecular species emerged during the reaction process is well supported by comprehensive characterizations. The manuscript is well written, and the data presented are of high quality. However, several major concerns as listed below prevent me from recommending its publication in Nat. Commun.

1) The main point of this work is introduction of a novel synthesis strategy for enetriynes with high selectivity. However, its universality as a new strategy is not confirmed, which significantly reduces its novelty and applicability. Does the selective synthesis of enetriyne only work for the specific precursor? Is it applicable for different on-surface systems? Experimental evidence on the universality of the synthesis strategy should be provided.

2) The authors have tried to provide detailed insights into the reaction process. Unfortunately, their explanations for the reaction mechanism, including the effects of the hydroxyl group, O₂ and substrate lattice, are quite ambiguous. i) How does the hydroxyl group affect the reaction selectivity? The authors state that the hydroxyl group plays a key role in the formation of the alternative dimer arrays on which the identified reaction mechanism is based. However, this statement is quite confusing because remarkable differences lie in between the structure of the alternative dimer arrays (Figure 2a) and the initial molecular configuration of the proposed reaction mechanism (Figure 5). Furthermore, direct evidence should be provided to compare the reactivities of the precursors with and without the hydroxyl group in order to address the role of the hydroxyl group. ii) To elucidate the role of O₂, the authors have carried out the experiment without O₂ dosage and observed the formation of disordered reaction products. However, no explanation for the underlying mechanism is provided. Is it due to the absence of the alkynyl-silver-alkynyl intermediates or incomplete dehydrogenation of the hydroxyl groups (as demonstrated by the authors, O₂ takes effect in two aspects, i.e., to facilitate the formation of the alkynyl-silver-alkynyl species and to promote the complete dehydrogenation of the hydroxyl groups)? iii) To provide a more comprehensive picture of the directing effect of the substrate lattice, additional control experiments concerning the reactivity of EHBP on other crystal surfaces like Ag(110) should be conducted.

3) In addition, there are two minor issues the authors should address. a) The authors identify a D_{4h} symmetry for the tetramer of the intact precursors, and a D_{2h} symmetry for the organometallic dimers. A D_{nh} symmetry requires a horizontal image plane at right angles to the n-fold principal axis. This implies an in-plane configuration of the molecular structure, which is not necessarily applicable to the tetramers or dimers in this work due to the existence of the substrate. b) The electronic structure of

enetriynes seems to be quite interesting with a conjugated molecular backbone. Have the authors experimentally explored the electronic properties of the enetriyne?

Reviewer #2 (Remarks to the Author):

Yang et al successfully developed a novel strategy to afford enetriyne with high chemoselectivity from EHBP on Ag(100) surface by control the temperature of the surface and the insertion of oxygen. Each step of the reaction was carefully measured by STM and nc-AFM measurements and the images were compared with the DFT-optimized structures. The reaction was also followed by XPS measurement to analyze the reaction mechanism in detail. Interestingly the four monomers reacted stepwise by changing the mutual orientation on Ag(100) to give enetriyne as 85% yield. Totally, the synthesis and the reaction mechanism of enetriyne on Ag(100) is worth publishing in Nature Communications after the revision of the following points.

1. The change from the structure in Figures 1b to 1d is drastic. Half of the molecules rotate 180°. What is a cause of the change? How did the dehydrogenation occur? What is the driving force in the change from Figure 1d to Figure 2?
2. By the O₂-mediated oxidization, how the Ag adatom is connected to the end of alkynyl skelton? Is there any difference between Ag(100) and Ag(111)? Addition reaction on Ag(111) was not effectively controlled (Figure S16). Please explain it. The reason the authors use the Ag(100) as the substrate should be mentioned.
3. Please explain how to decide the yield "85%" experimentally. Please show the STM image in larger scale before and after the reaction. About the byproducts, can you control the yield of the byproduct to increase the yield of enetriyne?
4. In Figure 5, Ag adatom moves at the final step. What is the driving force of the move?
5. Please describe the purity of EHBT.

Reviewer #3 (Remarks to the Author):

This is an interesting study which certainly contributes to the growing literature of "on surface synthesis".

The idea is original and the results are well articulated.

I have two concerns in relation to this study:

- even though the authors cite an extensive bibliography, they gloss over (too) many pioneering studies of on surface synthesis. Quoting the review by Clair and de Oteyza and a handful of other papers does

not cut it. Rather the authors should refer to a few of the early studies and to some recent milestone papers (that in fact appeared in Nature family journals within the past two years) to give proper credit and provide a suitable context.

- the conclusions are too dry and the outlook is insufficient.

Reviewer #4 (Remarks to the Author):

In this paper, authors report the on-surface synthesis of enetriynes. By combining STM, XPS, and DFT simulations, the structural features, bonding characteristics and underlying reaction mechanism are discussed. The study was well performed and results are reliable. My concerns mainly comes from the theoretical simulations.

In general, on-surface synthesis sensitively depends on the temperature of substrate. What is the temperature in simulation, room temperature?

The simulated reaction pathway indicate that the energy barrier for the synthesis ranges from 0.49 eV to 1.41 eV. However, the system is anneal at room temperature. It seems that the temperature is not high enough to overcome the calculated energy barrier.

Point-by-point response to the reviewers' comments (NCOMMS-22-29938)

Reviewer 1:

Cao and coauthors reported the selective synthesis of an enetriyne on Ag(100) by combined STM, AFM, XPS and DFT studies. The authors claim that the high selectivity stems from the introduction of a directing hydroxyl group, the formation of the alkynyl-silver-alkynyl organometallic intermediate species facilitated by the dosage of O₂, and the directing effect of the Ag(100) substrate lattice as well. The identification of the molecular species emerged during the reaction process is well supported by comprehensive characterizations. The manuscript is well written, and the data presented are of high quality. However, several major concerns as listed below prevent me from recommending its publication in Nat. Commun.

Reply: We appreciate the reviewer's positive and insightful comments. We have studied comments carefully and implemented, also based on extensive further investigations, a series of amendments that are expected to alleviate the raised concerns and enhance our study considerably.

(1) Comment: The main point of this work is introduction of a novel synthesis strategy for enetriynes with high selectivity. However, its universality as a new strategy is not confirmed, which significantly reduces its novelty and applicability. Does the selective synthesis of enetriyne only work for the specific precursor? Is it applicable for different on-surface systems? Experimental evidence on the universality of the synthesis strategy should be provided.

Reply: We appreciate the referee's feedback and agree that the applicability of our synthesis strategy to only a specific precursor may be seen as a limitation. However, we are confident in the potential of our integrated synthesis protocol, which is now analyzed in more depth and underpinned by a deep understanding of the mechanistic details, based on the comparative assessment of other precursors and refined modelization efforts. Accordingly, the introduced strategy opens novel avenues to systematically design precursors or adapted fabrication protocols for other cases, expanding the significance of our work.

There are three applied innovative aspects of our work, that is, introduction of a directing hydroxyl group to the aryl-alkyne molecular system, mediation of O₂ exposure and the application of a four-fold symmetric Ag(100) substrate. Each step is understood and can thus be transferred to other systems:

- a) The directing effect of hydroxyl group is based on its ability of forming strong hydrogen bonds between molecules¹, even including terminal oxygen residues after catalytic dehydrogenation on metal surfaces^{2,3}. Generally, they entail molecular nanostructures or layers in the form of ordered assembly structures, either for reactants, intermediates or products. This assisted-preassembly property not only provides favorable *in situ* reaction configurations, but also ensures product arrangements into pure domains, which is conducive to the stability and separation of the products^{3,4}.
- b) The O₂ exposure strategy was recently demonstrated for high-quality nanoarchitectures such as mesoscopically ordered alkynyl-silver networks⁵. The introduction of dioxygen brings a dramatic change in the reaction pathway of terminal alkynes⁶, extending the

reaction diversity of aryl-alkynes. It is worth mentioning that complete dehydrogenation of hydroxyl group by O₂ exposure has also been reported⁷, without affecting the reaction behavior of other functional groups.

- c) By using the Ag(100) substrate we capitalize on two advantages. First, the silver substrate is a proven suitable metal support for O₂-promoted convergent synthesis of alkynyl-silver complexes^{5,6}. Second, the four-fold symmetric surface lattice is widely applied to steer the formation of tetrameric units, including metal-organic coordination complexes^{3,8} and tetramerization products⁹.

The above understanding of the universal rules of each strategy is substantiated now by a series of careful control experiments. None of the attempts with simplified species yielded the enetriyne tetramers obtained by the above integrated strategy, including: i, using aryl-alkyne precursor without a hydroxyl group (4-ethynylbiphenyl and 4,4'-diethynylbiphenyl, details see **Figure S17** and **S18** in Supporting Information (SI) and **i) reply to Comment 2**); ii, direct thermal activation of the as-deposited molecular layers without O₂ exposure (details see **Figure S19** in SI and **ii) reply to Comment 2**); iii, probing the reactivity of EHBP on the hexagonal Ag(111) (details see **Figure S20** in SI and **iii) reply to Comment 2**).

- (2) **Comment:** *The authors have tried to provide detailed insights into the reaction process. Unfortunately, their explanations for the reaction mechanism, including the effects of the hydroxyl group, O₂ and substrate lattice, are quite ambiguous.*

i) How does the hydroxyl group affect the reaction selectivity? The authors state that the hydroxyl group plays a key role in the formation of the alternative dimer arrays on which the identified reaction mechanism is based. However, this statement is quite confusing because remarkable differences lie in between the structure of the alternative dimer arrays (Figure 2a) and the initial molecular configuration of the proposed reaction mechanism (Figure 5). Furthermore, direct evidence should be provided to compare the reactivities of the precursors with and without the hydroxyl group in order to address the role of the hydroxyl group.

Reply: We thank the referee for addressing the reaction mechanism and control experiments. Accordingly, we refined the pertaining descriptions and performed additional control experiments to complete the explanations and corroborate our understanding as well as data interpretation.

The hydroxyl group, as a directing group attached to the other end of the aryl-alkynes, plays the directing effect in the formation of ordered dimer arrays after O₂ exposure. The oxygen termini interact with biphenyl backbones through hydrogen bonds, stabilizing the organometallic Ag-bis-acetylide dimers in the array structures.

This O₂-mediated organometallic coupling was introduced in previous work^{5,6}. The final phase transformation in the experiment occurs upon thermally triggering the coupling of neighboring dimers, resulting in the tetramer assembly structures. Thus we further scrutinize the reaction mechanism from the alternative dimer arrays (**Figure 2a**). However, the presented IS configuration in **Figure 5** is not the starting point of our identified reaction mechanism. Rather, we carefully inspect the intermediate states (**Figure S12** in SI) and incomplete products (**Figure S9**) from the stepwise annealing experiments. The observed broken organometallic dimers, irregular trimeric and tetrameric species lead us to propose a scenario involving

stepwise organometallic bond scission and associated addition reactions, that is, the putative alternative reaction pathways shown in **Figure S13**. The DFT calculation for the cleavage of an organometallic dimer suggests pathway 1 to be the most favorable, whereby the incorporated Ag atom is fully removed from alkynyl groups. Thus an initial state of pathway 1 is assumed for the tetramerization with one organometallic dimer and two adjacent free monomers connected to the surface (IS in **Figure 5**).

Following to referee's suggestion, we provide additional evidence by studying the reactivity of two other precursors without the hydroxyl group, to address the role of the hydroxyl group. We first studied the reaction of 4-ethynylbiphenyl (EBP) on Ag(100). As shown in **Figure R1** (details see **Figure S17** in SI), without the interplay between hydroxyls, there are no regular structures found on the sample, neither as-deposited or following O₂ exposure. As a consequence, also the non-uniform addition products are randomly distributed on the surface after further thermal treatment.

Figure R1. Reaction scenario of EBP on Ag(100)

In addition, the precursor 4,4'-diethynylbiphenyl (DEBP; i.e., replacement of the hydroxyl with another alkynyl group), was studied for comparison with EHPB. As shown in **Figure R2** (details see **Figure S18** in SI), the self-assembly after initial deposition and organometallic chain structures obtained upon O₂ exposure reproduce results reported recently⁶. Due to the reactivity of the alkynyl groups at both ends and the absence of hydroxyl directing groups, the products are very disordered and pure enetriyne connections rarely express.

Figure R2. Reaction scenario of DEBP on Ag(100)

In conclusion, the hydroxyl groups play a crucial role in the molecular assemblies via tuning the interactions between adjacent molecules and the substrate, thus promoting the high selectivity of the reaction. Evidence is also provided by using two precursors without the hydroxyl group, directly revealing the relevance of the hydroxyl group. The complementary experimental data (Figure S17 and Figure S18) are added in the SI and mentioned in the main manuscript.

ii) To elucidate the role of O₂, the authors have carried out the experiment without O₂ dosage and observed the formation of disordered reaction products. However, no explanation for the underlying mechanism is provided. Is it due to the absence of the alkynyl-silver-alkynyl intermediates or incomplete dehydrogenation of the hydroxyl groups (as demonstrated by the authors, O₂ takes effect in two aspects, i.e., to facilitate the formation of the alkynyl-silver-alkynyl species and to promote the complete dehydrogenation of the hydroxyl groups)?

Reply: Thanks for the comments. As illustrated in Figure S19, annealing in the absence of oxygen led to the formation of disordered reaction products. During the review of the current manuscript we became aware of a systematic study of O₂ exposure effect⁶, which is now cited in the new version. It shows that molecular oxygen significantly reduces the activation energy of dehydrogenation of terminal alkynes⁶ and it is expected that there will be the same effect of the hydroxyl group⁷. Experimentally, the room temperature phase of EHBP deposited on silver surfaces reveals partially dehydrogenated hydroxyl constituents. As verified by AFM and XPS data (Figure 2 and 4), nearly complete dehydrogenation of hydroxyl is achieved only with O₂ dosage. Therefore, direct annealing the sample without O₂-exposure process will definitely induce a series of uncontrollable reactions because of the absence of the alkynyl-silver-alkynyl

intermediates and incomplete dehydrogenation of the hydroxyl groups. These possibilities include, but are not limited to C–C coupled enyne connections by the addition reaction of terminal alkynyl groups, C–O coupled enoether connection between terminal alkyne and hydroxyl groups. We added these proposed structures of side products to **Figure S19** in SI, as well as **Figure R3** in this reply letter.

Figure R3. Comparative experiment of annealing an EHBP layer on Ag(100) without previous O₂ exposure.

iii) To provide a more comprehensive picture of the directing effect of the substrate lattice, additional control experiments concerning the reactivity of EHBP on other crystal surfaces like Ag(110) should be conducted.

Reply: Thanks for the comment. To illustrate the directing effect of the substrate lattice, we report comparative experiments on Ag(111) surface with EHBP, as shown in **Figure S20** in SI (also in **Figure R4**). While the O₂ dosage on EHBP/Ag(111) sample induces the deprotonation of alkynyl groups and the formation of organometallic dimers, the condensed aggregation structures cannot afford effective control for the thermally triggered addition reaction upon further annealing. This result indicates that the Ag(100) substrate symmetry plays an important role for directing the addition reaction affording enetriynes. Unfortunately, we did not have a well-prepared Ag(110) sample at our disposal.

Figure R4. Reaction scenario of EHBP on Ag(111).

(3) **Comment:** In addition, there are two minor issues the authors should address.

a) The authors identify a D_{4h} symmetry for the tetramer of the intact precursors, and a D_{2h} symmetry for the organometallic dimers. A D_{nh} symmetry requires a horizontal image plane at right angles to the n -fold principal axis. This implies an in-plane configuration of the molecular structure, which is not necessarily applicable to the tetramers or dimers in this work due to the existence of the substrate.

Reply: Thanks for pointing out this shortcoming, which we sincerely regret. We accordingly adapted the description when referring to the symmetry of the tetrameric assemblies and organometallic dimers.

b) The electronic structure of enetriynes seems to be quite interesting with a conjugated molecular backbone. Have the authors experimentally explored the electronic properties of the enetriyne?

Reply: Thanks for the comment, clearly emphasizing an important aspect. The electronic properties of the enetriyne products were studied by scanning tunneling spectroscopy (STS) measurements and theoretical calculations, as shown in Figure R5.

Figure R5. Electronic structure of the enetriyne tetramer products.

We thus added these findings to the main text as Figure 6 with a paragraph of discussion: Page 12, paragraph 1: “As highly conjugated electron-rich compounds, enetriynes are associated with promising electronic properties. Theoretical calculation for the density of states (DOS) were performed based on the optimized structural model of an enetriyne species adsorbed on Ag(100) (cf. Figure 3d). We plotted the partial DOS (PDOS) and distinguished multiple pronounced orbitals which are attributed to the separate moieties of the molecular structure. As shown in Figure 6a, the most noticeable peak centered at 0.5 V represents the contribution from the enetriyne junction. With the conjugation of enetriyne in between, the HOMO-LUMO gap of the aromatic phenyl backbones is close to ~ 2.0 eV. To get insight into the local electronic properties of the enetriyne tetrameric units, we performed STS differential

conductance measurements (dI/dV) on the center of a tetramer (Figure 6b), and identified one predominant peak (-2.1 V) within a large energy window. With a closer inspection of the [-1.8 V, 1.2 V] bias range, several peaks referring to molecular orbitals are disentangled as compared to the reference spectrum of clean Ag(100) (Figure 6c). Without observing the state near 0 V, a derived energy gap of 1.8 eV is close to the calculated gap of aromatic backbones. The reasons for this might be of the large energy gap of the biphenyl backbones through connection to the conjugated enetriyne cores, as well as electronic hybridization with the substrate.”

Reviewer 2:

Yang et al successfully developed a novel strategy to afford enetriyne with high chemoselectivity from EHBP on Ag(100) surface by control the temperature of the surface and the insertion of oxygen. Each step of the reaction was carefully measured by STM and nc-AFM measurements and the images were compared with the DFT-optimized structures. The reaction was also followed by XPS measurement to analyze the reaction mechanism in detail. Interestingly the four monomers reacted stepwise by changing the mutual orientation on Ag(100) to give enetriyne as 85% yield. Totally, the synthesis and the reaction mechanism of enetriyne on Ag(100) is worth publishing in Nature Communications after the revision of the following points.

Reply: We highly appreciate the reviewer’s positive and constructive comments, as well as the recognition of our study's significance. Below please find our point-by-point response, including the corrections implemented in the main text.

(1) **Comment:** *The change from the structure in Figures 1b to 1d is drastic. Half of the molecules rotate 180°. What is a cause of the change? How did the dehydrogenation occur? What is the driving force in the change from Figure 1d to Figure 2?*

Reply: We thank the referee for the helpful comments. **Figure 1b and 1d** illustrate the respective units of the self-assembly structures at 150 K and room temperature (RT). The change of molecular orientation is caused by the evolution of intermolecular interactions via chemical changes induced by the substrate temperature. The molecules exhibit sufficient mobility on the surface to aggregate into the respective preferred structures, which generally applies to molecular systems with either hydroxyl^{2,3} or alkynyl^{10,11} terminals.

At 150 K, both terminal hydroxyl and alkynyl groups maintain intact on silver surfaces. The stronger intermolecular hydrogen bonding between four hydroxyl heads leads to the aggregation of tetramer clusters. Because of the directional nature of the hydrogen bond, the four equal monomers are perpendicular to each other. Upon annealing the sample to RT, parts of the hydroxyl groups dehydrogenate with the catalysis of silver substrate, which is supported by previous studies^{2,3} and the XPS measurements in the present study (**Figure S1**). Consequently, a new integrated supramolecular bonding motif is inferred, wherein C-H···O and O-H···O mixed hydrogen bonds form between alkyne entities and (dehydrogenated) hydroxyl species. Furthermore, the resulted large assembly networks reduce the average molecular adsorption energy, comparing to the former cluster-like adsorption structures at low temperature.

Similarly, exposing the assembly networks to O₂ deprotonates terminal alkyne moieties, forming Ag-bis-acetylide motifs within two monomers incorporating Ag adatoms from the silver substrate^{5,6}. The stronger alkynyl-Ag-alkynyl linkages make the organometallic dimers to be the dominant basic units. In addition, O₂ exposure also induces further dehydrogenation of hydroxyl groups^{4,7}, arranging dimers to extended array domains through the hydrogen bonds between oxygen termini and biphenyl backbones.

We've adapt and added the discussion of these phase transitions in the respectively paragraphs in the main text:

Page 5, paragraph 1: "...The drastic change of molecular orientation (from Figure 1b to 1d) is caused by the evolution of intermolecular interactions via chemical changes, since molecules exhibit sufficient mobility on the surface to aggregate into the respective preferred structures^{60,65,68}..."

Page 5, paragraph 2: "...To better characterize this phase transition... We thus hypothesize that the organometallic dimers have both hydroxyl groups dehydrogenated, which assignment is corroborated by XPS measurements (vide infra), arranging dimers to extended array domains through the hydrogen bonds between oxygen termini and biphenyl backbones."

- (2) **Comment:** *By the O₂-mediated oxidization, how the Ag adatom is connected to the end of alkynyl skeleton? Is there any difference between Ag(100) and Ag(111)? Addition reaction on Ag(111) was not effectively controlled (Figure S16). Please explain it. The reason the authors use the Ag(100) as the substrate should be mentioned.*

Reply: It also should be noted that the effect of dioxygen exposure is not an oxidation in the conventional sense. Considering that the presence of molecular oxygen to a large extent removes the barrier of dehydrogenation⁶ the plausible explanation is that the dehydrogenation happens prior to the connection between molecules and Ag adatoms. Similar to the Ag(111) surface, the coupling with an Ag adatom on Ag(100) will be exothermic. The remaining question is what drives the formation of organometallic *bis*-acetylides instead of C-C homocoupling. This would be of interest for several types of on-surface coupling reactions (including the on-surface Ullmann coupling), and is an intriguing question which we consider too complex to conclusively answer within the scope of the present study.

A significant difference of the O₂-mediated reaction protocol on Ag(111) is given by the arrangement of the resulting organometallic dimers, which are densely packed and oriented directionally in a given domain (Figure S20). This is concomitant with the direct interaction of O-termini, limiting the degree of freedom of molecules to participate in the following addition reaction. Thus the products were not effectively controlled.

The four-fold symmetric Ag(100) substrate not only directs the arrangement of organometallic intermediates, but also accords with the symmetry requirement of the tetramer product. Therefore it represents a favorable substrate.

- (3) **Comment:** *Please explain how to decide the yield "85%" experimentally. Please show the STM image in larger scale before and after the reaction. About the byproducts, can you control the yield of the byproduct to increase the yield of enetriyne?*

Reply: We thank the referee for the nice comments. The yield “85%” is determined by counting and identifying more than 2000 products from the high resolution large-scale STM images. The yield is then simply defined as the proportion of enetriynes to all products obtained on the sample.

Another possible definition of the yield would be the proportion of monomers for forming the enetriynes to all monomer reactants. We have now also estimated the yield in this way and came up with a value of 84%, similar to the result obtained for the initial definition. The new analysis is shown in **Figure R6** (and **Figure S10** in SI) with large scale STM images before and after the reaction.

Figure R6. Statistical analysis of enyne reaction products.

The initially deposited monomers transformed to organometallic dimers with nearly 100% yield, as long as the O₂ exposure is adequate. Losses in the final product yield are mainly due to the covalently coupled side products, such as deformed tetramers and trimeric species. These are mostly found at the edges of the domains. Hence we may obtain the higher yield by increasing the molecular coverage and O₂ exposure amount. We accordingly modified **Figure S9 and S10** in SI and also added the sentence in the main text:

Page 8, paragraph 2: “Importantly, although there are some molecular side products or spurious elements, such as trimers or deformed tetramers (Figure S9), the chemoselectivity towards the enetriyne is high (~85%), as deduced from a statistical analysis of extended STM data sets

(Figure S10). Note that side products are mostly found at the edges of the domains. Hence we may obtain the higher yield by increasing the molecular coverage and O₂ exposure.”

(4) **Comment:** *In Figure 5, Ag atom moves at the final step. What is the driving force of the move?*

Reply: We thank the referee for the stimulating remark. In the proposed reaction pathway from organometallic dimers to enetriyne tetramers, stepwise organometallic bond scission and associated addition reactions are involved. During the last step, going from InS3 to FS, the final C-C bond is formed simultaneously as the Ag atom is released from the complex. First of all, this step is exothermic with a reaction energy of -0.65 eV. In the transition state of this process (TS4), the silver atom is still chemically bonded to the complex at a different position, as the final C-C bond is formed. The energy gain going from TS4 to FS, together with the close distance of the Ag atom with two phenyl groups, qualitatively explains that the release of the Ag atom is driven by steric hindrance between the Ag atom and the organic complex, although it is difficult to quantify such a statement. Notably, going from TS4 to FS is a spontaneous process.

(5) **Comment:** Please describe the purity of EHBP.

Reply: It was purchased from AmBeed Company with a nominal purity of 95+%. We've modified the description in the Methods Section.

Reviewer 3:

This is an interesting study which certainly contributes to the growing literature of "on surface synthesis". The idea is original and the results are well articulated. I have two concerns in relation to this study:

Reply: We appreciate the positive comments from the reviewer, as well as the recognition of our study's significance.

(1) **Comment:** *Even though the authors cite an extensive bibliography, they gloss over (too) many pioneering studies of on surface synthesis. Quoting the review by Clair and de Oteyza and a handful of other papers does not cut it. Rather the authors should refer to a few of the early studies and to some recent milestone papers (that in fact appeared in Nature family journals within the past two years) to give proper credit and provide a suitable context.*

Reply: Thanks for the comments. We agree with the referee that the citations of the on-surface synthesis concept can be improved here. We've modified and cited more representative original papers from early and recent studies:

Page 2, paragraph 3: “On-surface synthesis^{20,21} has introduced alternative routes towards the formation of novel conjugated nanostructures²²⁻²⁴ and functional organic molecules²⁵⁻²⁸. The confinement and catalytic activity on surfaces promote the transformation of reactive groups, enabling the formation of distinct organic compounds or nanostructures^{29,30}. The reinforcing of bond-resolved non-contact atomic force microscopy (nc-AFM)^{31,32}, recently provides access to the precise fabrication of covalent organic nanoarchitectures³³⁻³⁶.”

(2) **Comment:** *The conclusions are too dry and the outlook is insufficient.*

Reply: We agree with the referee regarding this point. We have modified and polished the conclusions in the new version and included possible future research directions:

Page 12, paragraph 2: “...The electronic properties of the enetriyne tetramers were studied by STS measurements and PDOS calculations, signaling a high conjugation of the enetriyne junction and connected aromatic ring systems. The present integrated synthetic approach notably combines a general precursor design and customized experimental preparation protocols based on the recognized properties of molecular substituents as well as the substrate symmetry and catalytic activity. Our study introduces a high-yield procedure for the *in vacuo* preparation of enetriyne derivatives on surfaces. We envision that the gained knowledge and methodological insights can be further developed towards the precise synthesis of molecular wires or reticulated two-dimensional networks featuring embedded enetriyne units, affording novel functionalities and bearing application potential.”

Reviewer 4:

In this paper, authors report the on-surface synthesis of enetriynes. By combining STM, XPS, and DFT simulations, the structural features, bonding characteristics and underlying reaction mechanism are discussed. The study was well performed and results are reliable. My concerns mainly comes from the theoretical simulations.

Reply: We appreciate the reviewer’s positive comments and appraisal of our study. Complementing the enlarged scope of experimental work, we address the referee’s concerns about the theoretical calculations as follows:

(1) **Comment:** *In general, on-surface synthesis sensitively depends on the temperature of substrate. What is the temperature in simulation, room temperature?*

Reply: We thank the referee for spotting this issue. The calculations consider the potential energy landscape at 0 K, which is a standard procedure when studying reactions on surfaces. We are mainly interested in the activation energies of the different reaction steps. These are indeed sensitive to temperature, to some extent, since vibrational energies are affected throughout a reaction. In particular since transition states have one vibrational degrees of freedom less than local minima, resulting in lowering of barriers. However, previous studies have demonstrated this to be at the order of 0.1 eV¹² and would not affect the conclusions.

For a better understanding, addition calculations for the rate-limiting step (InS3 to FS) of the overall reaction including vibrational entropy and enthalpy (520 K) were performed. Interestingly, the results reveal a rate-limiting barrier affected by temperature. Essentially, the barrier becomes slightly higher (0.11 eV) (see **Figure R7**), which does not pose restrictions for the experimental reaction conditions. The new calculations showing the effect of temperature were included in the revised Supporting Information (**Figure S16**).

Figure R7. Calculated energy profiles of the last step of the overall reaction, comparing the electronic enthalpy (ΔH^{elec}) to the free energy (ΔG) at 520 K. The free energy was calculated within the harmonic approximation, adding vibrational enthalpy and entropy to the electronic enthalpy.

(2) **Comment:** The simulated reaction pathway indicate that the energy barrier for the synthesis ranges from 0.49 eV to 1.41 eV. However, the system is anneal at room temperature. It seems that the temperature is not high enough to overcome the calculated energy barrier.

Reply: Experimentally the system is indeed annealed up to 520 K (room temperature conditions are merely used for intermediate steps) to overcome the reaction barrier. According to the Eyring equation, a barrier of 1.41 eV at a temperature of 520 K gives a reaction rate of 0.2 s^{-1} (and a rate of 0.02 s^{-1} for the free energy barrier of 1.52 eV - assuming the standard pre-factor in the Eyring equation of $\frac{k_B T}{h}$). I.e., the temperature in the experiments are clearly sufficient to overcome the barriers suggested by theory.

Reference

1. Marele, A. C., *et al.* Some pictures of alcoholic dancing: From simple to complex hydrogen-bonded networks based on polyalcohols. *J. Phys. Chem. C* **117**, 4680-4690 (2013).
2. Feng, L., *et al.* Supramolecular tessellations at surfaces by vertex design. *ACS Nano* **13**, 10603-10611 (2019).
3. Yang, B., *et al.* Intermediate states directed chiral transfer on a silver surface. *J. Am. Chem. Soc.* **141**, 168-174 (2019).
4. De Marchi, F., *et al.* Room-temperature surface-assisted reactivity of a melanin precursor: Silver metal–organic coordination versus covalent dimerization on gold. *Nanoscale* **10**, 16721-16729 (2018).
5. Zhang, Y.-Q., *et al.* Synthesizing highly regular single-layer alkynyl–silver networks at the micrometer scale via gas-mediated surface reaction. *J. Am. Chem. Soc.* **141**, 5087-5091 (2019).
6. Zhang, C., Kazuma, E., Kim, Y. Steering the reaction pathways of terminal alkynes by introducing oxygen species: From C–C coupling to C–H activation. *J. Am. Chem. Soc.* **144**, 10282-10290 (2022).
7. De Marchi, F., *et al.* Self-assembly of 5,6-dihydroxyindole-2-carboxylic acid: Polymorphism of a eumelanin building block on Au(111). *Nanoscale* **11**, 5422-5428 (2019).

8. Uphoff, M., *et al.* Assembly of robust holmium-directed 2D metal–organic coordination complexes and networks on the Ag(100) surface. *ACS Nano* **12**, 11552-11560 (2018).
9. Li, Q., *et al.* Self-assembly directed one-step synthesis of [4]radialene on Cu(100) surfaces. *Nat. Commun.* **9**, 3113 (2018).
10. Li, Q., *et al.* Supramolecular self-assembly of π -conjugated hydrocarbons via 2D cooperative CH/ π interaction. *ACS Nano* **6**, 566-572 (2011).
11. Wang, T., Lv, H., Fan, Q., Feng, L., Wu, X., Zhu, J. Highly selective synthesis of cis-enediynes on a Ag(111) surface. *Angew. Chem. Int. Ed.* **56**, 4762-4766 (2017).
12. Björk, J. Thermodynamics of an electrocyclic ring-closure reaction on Au(111). *J. Phys. Chem. C* **120**, 21716-21721 (2016).

REVIEWERS' COMMENTS

Reviewer #1 (Remarks to the Author):

After major revisions, the manuscript is now acceptable for publication in Nat. Commun. without further reviewing, subject to conduct the following minor revisions.

1) While they have carried out more experiments and/or added more descriptions to address my previous concerns, the authors should soften their tune of claims in the text and conclusion in terms of the anticipated applicability without solid experimental evidence. In specific, “We envision that ... can be further developed towards ...” in the conclusion part should be replaced by “We envision that ... might be further developed towards ...”. Don't be far-fetched and misleading in scientific conclusions without concrete experimental evidence.

2) In my previous comment 2-iii), “To provide a more comprehensive picture of the directing effect of the substrate lattice, the comparative study of the reaction of EHBP on Ag(110) should also be carried out.” The authors reply as: “This result indicates that the Ag(100) substrate symmetry plays an important role for directing the addition reaction affording enetriynes. Unfortunately, we did not have a well-prepared Ag(110) sample at our disposal.” Two aspects should be kept in mind. On the one hand, the four-fold symmetric Ag(100) substrate seems to be unique, which suggests once again that the reported strategy be not universal and widely applicable, against the authors' claim. On the other hand, the statement - “Unfortunately, we did not have a well-prepared Ag(110) sample at our disposal.” – cannot be simply used as an excuse to subjectively insist on their claim. One should be aware of that the surface lattice of Ag(110) is rectangular in unit cell, being somewhat similar to the square unit cell of the Ag(100) surface lattice. The right angles in both unit cells may dictate the formation of the tetramers or similar aggregates that are vital to mediate eventual products. Therefore, the authors should also tentatively clarify the directing effect of the underlying substrate lattice in relating parts.

Reviewer #2 (Remarks to the Author):

The authors replied the comments of the reviewers appropriately and revised the manuscript properly. The purity 95% of EHBP could be improved in future. The reviewer agreed for the publication of the manuscript as it is in Nature Communications.

Reviewer #3 (Remarks to the Author):

The authors have revised their manuscript in accordance with the criticisms and suggestions raised by previous reviewers. they have also provided extensive responses to all the concerns that were raised. At this point this manuscript can be accepted for publication in its current form.

Reviewer #4 (Remarks to the Author):

Authors have addressed my concerns on this paper. This paper is publishable in current format.

Point-by-point response to the reviewers' comments (NCOMMS-22-29938A)

Reviewer 1:

After major revisions, the manuscript is now acceptable for publication in Nat. Commun. without further reviewing, subject to conduct the following minor revisions.

Reply: We appreciate the referee for reviewing our manuscript again and providing positive feedback and insightful comments. We have studied comments carefully and conducted the minor revisions to enhance our study.

- (1) **Comment:** *While they have carried out more experiments and/or added more descriptions to address my previous concerns, the authors should soften their tune of claims in the text and conclusion in terms of the anticipated applicability without solid experimental evidence. In specific, “We envision that ... can be further developed towards ...” in the conclusion part should be replaced by “We envision that ... might be further developed towards ...”. Don't be far-fetched and misleading in scientific conclusions without concrete experimental evidence.*

Reply: We appreciate the referee's feedback and suggestion. We've polished some related expressions in the text.

Last sentence in the **Abstract** section, “Our study provides an integrated strategy for the precise fabrication of functional enetriyne species, thus providing access to a distinct class of highly conjugated π -system compounds.

Last sentence in the **Conclusion** section, “We envision that the gained knowledge and methodological insights may be further developed towards the precise synthesis of molecular wires or reticulated two-dimensional networks featuring embedded enetriyne units, affording novel functionalities and bearing application potential.”

- (2) **Comment:** *In my previous comment 2-iii), “To provide a more comprehensive picture of the directing effect of the substrate lattice, the comparative study of the reaction of EHBP on Ag(110) should also be carried out.” The authors reply as: “This result indicates that the Ag(100) substrate symmetry plays an important role for directing the addition reaction affording enetriynes. Unfortunately, we did not have a well-prepared Ag(110) sample at our disposal.” Two aspects should be kept in mind. On the one hand, the four-fold symmetric Ag(100) substrate seems to be unique, which suggests once again that the reported strategy be not universal and widely applicable, against the authors' claim. On the other hand, the statement - “Unfortunately, we did not have a well-prepared Ag(110) sample at our disposal.” – cannot be simply used as an excuse to subjectively insist on their claim. One should be aware of that the surface lattice of Ag(110) is rectangular in unit cell, being somewhat similar to the square unit cell of the Ag(100) surface lattice. The right angles in both unit cells may dictate the formation of the tetramers or similar aggregates that are vital to mediate eventual products. Therefore, the authors should also tentatively clarify the directing effect of the underlying substrate lattice in relating parts.*

Reply: We thank the referee for the kind criticism and constructive suggestion. In fact, the use of unique Ag(100) substrate is one of our designed strategies to afford enetriyne motifs based

on the principle of symmetric matching. Nonetheless, the reviewer's reminder is very enlightening for our further study. The Ag(110) surface with rectangular lattice might be also applicable to guide the arrangement of organometallic intermediates and afford final enetriyne products, improving the universal and widely applicable of our synthesis strategy.

In the **Abstract** section, we add “Taking advantage of a directing hydroxyl group, we steer molecular assembly and reaction processes **on square lattices.**”

In the **Discussion** section, we add “**Therefore, the square symmetry of Ag(100) surface lattice plays an important role in the aggregation of organometallic dimers and subsequently evolving enetriyne species.**”

Reviewer 2:

The authors replied the comments of the reviewers appropriately and revised the manuscript properly. The purity 95% of EHBP could be improved in future. The reviewer agreed for the publication of the manuscript as it is in Nature Communications.

Reply: We thank the reviewer for reviewing our manuscript again and providing the positive feedback. We are very grateful for those constructive comments guiding us to improve the quality of our work.

Reviewer 3:

The authors have revised their manuscript in accordance with the criticisms and suggestions raised by previous reviewers. They have also provided extensive responses to all the concerns that were raised. At this point this manuscript can be accepted for publication in its current form.

Reply: We thank the reviewer for reviewing our manuscript again and providing the positive feedback. We are very grateful for those constructive comments guiding us to improve the quality of our work.

Reviewer 4:

Authors have addressed my concerns on this paper. This paper is publishable in current format.

Reply: We thank the reviewer for reviewing our manuscript again and providing the positive feedback. We are very grateful for those constructive comments guiding us to improve the quality of our work.